# Intracoronary-Cardiosphere-Derived Cell Secretome Therapy: Effects on Ventricular Tachycardia Inducibility and Cardiac Function in a Swine Model

**DOI:** 10.3390/biomedicines13051043

**Published:** 2025-04-25

**Authors:** Claudia Báez-Díaz, Axiel Torrescusa-Bermejo, Francisco Miguel Sánchez-Margallo, Fátima Vázquez-López, María Pulido, Esther López, Ángel Arenal, Verónica Crisóstomo

**Affiliations:** 1Fundación Centro de Cirugía de Mínima Invasión Jesús Usón, 10071 Cáceres, Spain; atorrescusa@ccmijesususon.com (A.T.-B.); msanchez@ccmijesususon.com (F.M.S.-M.); fvazquez@ccmijesususon.com (F.V.-L.); mpulido@ccmijesususon.com (M.P.); elopez@ccmijesususon.com (E.L.); crisosto@ccmijesususon.com (V.C.); 2Red Española de Terapias Avanzadas RICORS-TERAV, 28029 Madrid, Spain; 3Hospital General Universitario Gregorio Marañón, 28007 Madrid, Spain; arenal@secardiologia.es

**Keywords:** S-CDCs, ventricular tachycardia, myocardial infarction, swine

## Abstract

**Background/Objectives:** Ventricular tachycardia (VT) resulting in sudden cardiac death is common following a myocardial infarction (MI). Our objective was to evaluate the effects of an intracoronary (IC) administration of cardiosphere-derived cell secretome (S-CDCs) on VT inducibility and cardiac function in a swine model of MI. **Methods:** Fourteen pigs underwent endovascular MI model creation. At 4 weeks, saline (CON; 5 mL; n = 7) or S-CDCs (S-CDCs; 9.16 mg protein in 5 mL saline; n = 7) was blindly administered via the IC route. VT inducibility and magnetic resonance imaging (MRI) studies were performed both pre- and 4 months post-IC therapy, calculating left ventricular ejection fraction (LVEF), infarct size as a percentage of left ventricle (% MI), and left ventricular indexed end-diastolic and end-systolic volumes (LVEDVi, LVESVi). **Results:** While VT was inducible in 100% of the animals before IC therapy, at 4 months, the inducibility rate was lower in the S-CDCs group compared to the CON group (57% versus 100%, *p* = 0.05). Likewise, in the S-CDCs group, % MI was significantly lower than in the CON group (12 ± 3% versus 16 ± 3%, *p* = 0.03). LVEF (S-CDCs: 35 ± 10% versus CON: 29 ± 10%, *p* = NS), LVEDVi and LVESVi (S-CDCs: 83 ± 18 mL/m^2^ and 56 ± 20 mL/m^2^ versus CON: 88 ± 29 mL/m^2^ and 64 ± 20 mL/m^2^, *p* = NS) did not change. **Conclusions:** IC therapy with S-CDCs appears to reduce the development of post-MI VT. Furthermore, it suggests a beneficial effect on infarct size, reducing % MI in this experimental swine model.

## 1. Introduction

Myocardial infarction (MI) carries a high risk of sudden cardiac death (SCD) primarily due to the development of ventricular tachycardia (VT) [1]. Specifically, almost 30% of the patients suffer from SCD within the first month, being responsible for up to 40% of post-MI mortality [2,3].

Post-infarction VT is categorized as early or late depending on the time of onset: Early VT, occurring within 48–72 h post-MI, is generally transient and associated with acute ischemia, reperfusion injury, or metabolic imbalances, with treatment aimed at correcting reversible factors. In contrast, late VT, developing after 72 h (often weeks to months later), is typically caused by re-entry circuits established by slow conduction channels within the heterogeneous tissue (HT) immersed in the non-excitable dense scar. VT occurring more than 40 days post-MI is generally considered permanent, as it reflects an underlying scar-related substrate. The management of late VT frequently involves implantable cardioverter defibrillators and catheter ablation, among others [4].

Catheter ablation is a common technique that is carried out in up to 20% of the patients for targeting this arrhythmogenic substrate [5]. Emerging therapeutic strategies are exploring the modulation of this process through biological agents, offering a promising frontier in the treatment of post-MI arrhythmias [6].

Among these emerging strategies, cardiosphere-derived cells (CDCs), a heterogeneous population of cardiac progenitor cells typically isolated from heart tissue biopsies, have shown significant promise in preclinical and clinical settings. These cells, generated through a multistep culture process involving enzymatic digestion, formation of cardiospheres, and subsequent adherent culture, are able to reduce scar mass, enhance viable myocardial tissue, and prevent adverse remodelling, thereby offering a potential way of halting the development of new arrhythmogenic substrates [7,8,9,10,11]. However, recent findings suggest that the therapeutic effects of CDCs are largely mediated by the release of their secretome (S-CDCs) [12,13].

The growing body of evidence surrounding S-CDCs underlines their potential as a key technological advancement in cardiac regenerative medicine. Preliminary studies have demonstrated that S-CDC therapy yields results comparable to those of the cells themselves, including a reduction in scar size and an increase in viable myocardial mass in chronic MI models [14,15]. These findings point to a promising new direction for drug development and therapeutic intervention in arrhythmogenic disorders, where the focus is shifting towards cell-free, biologically active compounds.

One of the key advantages of S-CDCs is their acellular nature, which avoids some of the challenges associated with cell therapy. As an acellular and non-replicating product, S-CDCs can be developed into a stable and reliable “off-the-shelf” therapy, greatly enhancing their potential for widespread clinical application [15]. This shift towards a cell-free approach also aligns with technological advances in drug delivery systems, biomaterials, and regenerative medicine, creating new opportunities for treating cardiac arrhythmias and related disorders.

Regarding the anti-arrhythmic properties of S-CDCs, its administration has been shown to reduce the inducibility of sustained ventricular arrhythmias, when administered transendocardially [1]. This demonstration of an anti-arrhythmic effect of S-CDCs motivates more extensive research of the effects of these particles on rhythm disorders. To this regard, we hypothesized that an IC administration of S-CDCs could achieve similar effects. However, it needed to be tested in a clinically relevant animal model. Thus, the objective of this study was to evaluate the effects of an IC administration of S-CDCs on VT inducibility and cardiac function in a porcine model of chronic MI.

## 2. Materials and Methods

The study protocol was approved by the Jesús Usón Minimally Invasive Surgery Centre Animal Care and Use Committee (Ref 017/20) and the Extremadura Regional Government (Exp 20201027), and it complied fully with the Directive 2010/63/EU of the European Parliament on the protection of animals used for scientific purposes. This animal study is reported in accordance with the ARRIVE Guidelines for reporting experiments involving animals.

Fourteen female Large White swine weighing 30–35 kg were subjected to infarct induction, using an MI model characterized by large heterogeneous scars with high VT inducibility rates [6]. Animals included in the study were blindly allocated (randomization was carried out by an independent researcher using random number generation) to one of the following groups: the control (CON) or S-CDCs group. The experimental protocol is summarized in Figure 1.

### 2.1. Anaesthesia and Analgesia

Animals were placed under general anaesthesia. For that purpose, swine were premedicated with 20 mg/kg intramuscular (I.M.) ketamine (Ketamidor 100 mg/mL, Richter Pharma AG, Wels, Austria). Fifteen minutes later, the induction of anaesthesia was accomplished with 3 mg/kg intravenous (I.V.) 1% propofol (Propofol-Lipuro; B Braun Ltd., Melsungen, Germany). Anaesthetic maintenance was achieved with inhaled sevoflorane (1.8–2% inspiratory fraction). Endotracheal tubes were connected to a semi-closed circular anaesthetic circuit attached to a ventilator (Maquet Flow i) with a fresh gas flow rate of 1 L/min (0.4/0.6 mixture of oxygen and air). Controlled ventilation was set up with a tidal volume of 10 mL/kg to obtain normocapnia (with a CO_2_ pressure of 40–45 mmHg). Continuous lidocaine (Lidocaína 2% Braun, B Braun Ltd., Melsungen, Germany) was administered at an infusion rate of 1 mg/kg/h. Anaesthetic monitoring was carried out by measurement of heart rate, electrocardiography, pulse-oximetry and invasive arterial blood pressure.

Postoperative analgesia included I.M. buprenorphine (10 μg/kg/12 h) during the first 24 hours combined with a fentanyl transdermic release patch (50 μg/h).

### 2.2. Infarct Induction

The protocol for infarct creation has been described elsewhere [6]. In brief, a percutaneous coronary angioplasty balloon (Ryujin plus PTCA dilatation catheter, Terumo, Tokyo, Japan) with an appropriate diameter was used to transiently occlude the left anterior descending coronary artery (distal to the second diagonal branch) via a percutaneous femoral approach. Balloon occlusion was maintained during 150 min followed by reperfusion. Possible ventricular fibrillation episodes during infarct induction were treated using manual chest compressions and 200 J biphasic defibrillation shocks (Zoll M series biphasic 200 J, Zoll Medical Corporation, Chelmsford MA, USA) as well as medication when needed.

### 2.3. Isolation of S-CDCs

S-CDCs was obtained from CDCs cultured at 80% confluence (passages 12–15). Culture medium was replaced by secretome isolation medium comprising 1% insulin–transferrin–selenium in DMEM supplemented with 10% Penicillin/Streptomycin.

Supernatants were collected on day four and subjected to two centrifugation steps at 4 °C: first at 1000× *g* for 10 min, followed by 5000× *g* for 20 min. In order to eliminate dead cells and debris, the supernatants were filtered through a 0.22 μM filter and then concentrated using a 3 kDa MWCO Amicon^®^ Ultra device (Merck-Millipore, Burlington, MA, USA) at 4000× *g* for 40 min at 4 °C. The protein concentration in the enriched S-CDCS was determined using the Bradford assay (Bio-Rad Laboratories, Hercules, CA, USA). Concentrates were recovered and stored at −80 °C until use.

### 2.4. IC Administration

Four weeks after infarct induction, the pigs were anesthetized again. A 3 Fr microcatheter (Microferret infusion catheter, Cook Medical, Letchworth, UK) was placed in the anterior descending coronary artery (just at the site where the occlusion had been performed) to blindly receive an IC administration of saline (CON; 5 mL; n = 7) or secretome (S-CDCs; 9.16 mg of protein in saline; 5 mL; n = 7) at an injection speed of 1 mL/minute. In order to evaluate the safety of the IC therapy, blood parameters (creatinin, glucose, GOT, GPT, CRP, total proteins, urea, troponin I and CK-MB) were determined both before and after IC administration as well as 4 months after it (AQT90 Flex, Radiometer Iberica SL, Madrid, Spain).

### 2.5. MRI Studies

MRI examinations were performed before and four months after IC injection, as previously described [16]. To that end, anesthetized pigs were positioned in sternal decubitus inside the MRI system (Intera 1.5 T, Philips Medical Systems, Best, The Netherlands). Short-axis breath-hold gradient-echo cine images were used to measure left ventricular function including end diastolic volume (LVEDV), end systolic volume (LVESV) and ejection fraction (LVEF). LVEDV and LVESV were normalized by body surface area (BSA) in order to allow comparison over time. Short-axis images were acquired 10 min after the injection of 0.2 mmol/kg of a gadolinium-based contrast agent (Gadobutrol. Gadovist 1.1 mmol/L, Bayer Schering Pharma AG, Berlin, Germany) using a breath-hold 3D gradient-echo inversion-recovery sequence to calculate infarct size (% MI). All MRI-derived measurements were performed by a blinded operator.

### 2.6. Programmed Electrical Stimulation (PES)

PES was carried out by means of a quadrapolar catheter (Marinr SC Steerable Quadrapolar Catheter, Medtronic, Minneapolis, MN, USA) inserted into the left and right ventricles in order to analyse the inducibility of arrhythmias before IC therapy (four weeks after MI creation) and four months later (after MRI examination). A clinical stimulation protocol based on the application of three cycle lengths with up to 4 extrastimuli with different coupling intervals was applied.

### 2.7. End Study

Once the MRI and VT inducibility studies at 4 months were completed, the animals were euthanized using a lethal dose of potassium chloride (1–2 mmol/kg). Thereafter, the hearts were explanted and sectioned into 0.5 cm-thick slices, with one of these sections incubated for 10 min in a 1% solution of 2,5,3-triphenyltetrazolium chloride (TTC) in phosphate buffer at 37 °C. Samples from normal tissue (NT, normal myocytes and the absence of fibrosis) and infarcted area were collected from one of the remaining slices for posterior histopathological analysis by means of haematoxylin–eosin (HE) and Masson’s trichrome (MT) staining. Macroscopic lesions (edema, haemorrhage, necrosis and inflammation) and the histology of the myocytes (hypertrophy, myocytolysis, edema, vacuolar degeneration) were analysed. The infarct zone was differentiated into two substructures: central dense scar (DS) and lateral heterogenous tissue (including the heterogenous tissue surrounding the dense scar, HT).

### 2.8. Statistical Analysis

The obtained data are presented as means ± standard deviations. Differences between groups were calculated using the Mann–Whitney U tests. The Wilcoxon paired sample test was used to perform intragroup comparisons (pre- and 4 months post-therapy timepoints). For repeated measures the Friedman test was performed. Binary data were analysed by performing a chi-square test. Values of *p* < 0.05 were considered statistically significant. All calculations were carried out using the SPSS 18.0 statistical package for Windows (SPSS Inc., Chicago, IL, USA).

## 3. Results

All 14 animals survived the induction of the MI model. Moreover, IC administration was carried out successfully in all swine in the absence of changes in ECG or adverse cardiac events.

Mean values of the studied blood-derived parameters are presented in Appendix A. Although intragroup comparisons revealed statistically significant differences in S-CDCs (GOT, GPT, CRP, Total Proteins and Troponin I) and in CON (GOT, CRP, Total Proteins, Urea, Troponin I and CK-MB), the mean values remained clinically normal in both groups during the complete follow-up period (creatinine and urea levels were slightly increased over the upper range reference level in the CON group before and after IC therapy timepoints (Appendix A). After therapy (immediately and at four months), no statistically significant differences between groups were detected in any measured variable.

While pre-therapy PES study showed inducible VT in all animals (both groups), VT inducibility at four months was lower in the S-CDCs (four out of seven pigs were inducible) compared to CON (all swine presented VT): 57% vs. 100%, *p* = 0.05 (Figure 2A).

Regarding the parameters evaluated by MRI (Table 1), before therapy, no significant differences between the CON and S-CDCs groups were observed in LVEF (32 ± 6% vs. 35 ± 5%, *p* = 0.30, respectively), % MI (14 ± 4 vs. 15 ± 4%, *p* = 0.56, respectively), LVEDVi (88 ± 19 mL/m^2^ vs. 83 ± 14 mL/m^2^, *p* = 0.52, respectively) and LVESVi (66 ± 15 mL/m^2^ vs. 58 ± 13 mL/m^2^, *p* = 0.10, respectively).

At 4 months post-therapy, animals in the S-CDCs group showed a trend towards higher LVEF than the swine belonging to CON group (35 ± 10% vs. 29 ± 10%, *p* = 0.25) (Figure 3A). On the contrary, LVESVi and LVEDVi were lower in the S-CDCs group compared to the CON group (56 ± 20 mL/m^2^ and 83 ± 18 mL/m^2^ vs. 64 ± 27 mL/m^2^ and 88 ± 29 mL/m^2^; *p* = 0.41 and *p* = 0.66, respectively) (Figure 3B,C), without reaching statistical significance in any case. Infarct size, however, was significantly lower in the group treated with S-CDCs (12 ± 3% vs. 16 ± 3%; *p* = 0.03) (Figure 2B). Changes over time in the different parameters were not significant in any case.

TTC-stained heart sections showed transmural fibrous scars of varying extent and anteroseptal location in all samples of both groups (Figure 4). Thinning of the ventricular wall as well as a mixture of infarcted and viable tissue was observed in all animals.

Regarding the analysis of macroscopic lesions, no significant differences in the severity of edema, haemorrhage, necrosis and inflammation were detected between groups for the different areas (DS, HT) that were evaluated. With respect to the histology of the myocytes, there were no significant differences between groups in the percentage of animals exhibiting hypertrophy, myocytolysis, edema and vacuolar degeneration. Appendix A shows macroscopic and histological analysis results of the DS area.

## 4. Discussion

In this randomized, blinded preclinical study, we aimed to evaluate the effects of an IC administration of S-CDCs compared to saline on VT inducibility and cardiac function in a porcine model of chronic MI. At the 4-month follow-up, our results revealed a significant reduction in VT inducibility in the S-CDCs treated group compared to the CON group. Furthermore, MRI analysis demonstrated an improvement in infarct size in the S-CDCs group at that timepoint, suggesting a beneficial therapeutic effect of S-CDCs on the damaged myocardial tissue.

Over the past few years, CDCs have emerged as a promising therapeutic strategy for ischemic heart disease, demonstrating regenerative potential and favourable outcomes in preclinical MI models [7,8,9,10,11,17]. Moreover, in a recent study, we demonstrated that CDCs could modify the structure and electrophysiology of the post-infarction scar, thus reducing the risk of VT development [6].

In our current study, we focused on investigating the therapeutic potential of the secretome of CDCs as an alternative to cell-based therapies. The interest in cell-free therapies has grown due to the potential for overcoming the challenges associated with direct cell delivery, such as cell survival, engraftment, and differentiation. S-CDCs are thought to mediate many of the regenerative effects observed in CDC-based therapies, including anti-inflammatory and pro-angiogenic properties, tissue remodelling, and resolution of fibrosis [12,13]. Preliminary studies suggest that S-CDCs can replicate many of the benefits of CDCs in terms of cardiac repair, thus opening the door to cell-free therapies for the treatment of MI and associated arrhythmias [14].

Our research group has previously fully characterized the secretome released by CDCs [18], and its immunomodulatory effects have been examined in a porcine MI model [17]. However, these studies were not focused on the potential effectiveness of S-CDCs as antiarrhythmic and/or MI therapy [19].

The choice of animal model is a critical consideration in evaluating the potential of any new therapeutic strategy [20]. In our case, we utilized a porcine MI model induced by 150 min of balloon occlusion. This model is widely regarded for its ability to reliably induce VT, with 90% of animals typically demonstrating VT inducibility within 4–5 weeks post-MI [21]. In our study, we achieved a 100% inducibility rate at the 4-week follow-up, which aligns with previously published data from similar models [22]. This high inducibility rate reflects the effectiveness of the used model in reproducing the electrophysiological substrate for VT, providing an appropriate platform to assess the antiarrhythmic effects of S-CDCs.

The pathophysiology of post-MI VT is closely linked to the presence of re-entry circuits within the infarcted tissue. These circuits typically occur in areas of slow conduction HT, where myocyte bundles are embedded within the fibrotic tissue. Abnormal electrograms and late potentials are characteristic features of these regions, which serve as the arrhythmic substrate for VT. The size and electrophysiological properties of the HT significantly influence VT inducibility, and any therapeutic intervention that can modulate these properties may reduce the risk of arrhythmia [6,22]. In the present study, we hypothesized that S-CDCs could help modify the electrophysiological properties of the infarcted tissue, thereby reducing the substrate for VT.

Despite advancements in medical therapies, including antiarrhythmic drugs, autonomic modulation, implantable cardioverter defibrillators (ICDs), and catheter ablation, current treatments for post-MI VT remain suboptimal [23]. Biological treatment options, including cell-based therapies, have shown promise in modulating the VT substrate and reducing arrhythmia inducibility, but their clinical applicability remains limited [24].

Recently, transendocardial therapy with S-CDCs has been shown to reduce the inducibility of sustained ventricular arrhythmias [1]. In the present study, we tried to clarify whether the IC delivery of S-CDCs could achieve similar effects as its transendocardial administration. To the best of our knowledge, we were able to evidence, for the first time, a lower inducibility rate in animals treated with S-CDCs by the IC route. As in the case of transendocardial delivery, we assume that VT is rendered non-inducible by the S-CDCs via improving conduction in areas with isolated potentials [1].

From a clinical point of view, the IC route presents several advantages over transendocardial therapy: on the one hand, it eliminates the need for specific equipment and/or training to perform this type of intervention, and on the other, it achieves a homogenous distribution of the delivered therapeutic agent. Nonetheless, IC administration has been associated with the appearance of intimal dissection [25], microvascular obstruction or even infarction [7,26,27]. To this regard, in this work, we analysed the safety of the IC therapy of S-CDCs by the appearance of alterations in the ECG and blood-derived parameters. Neither changes in ECG nor clinically relevant alterations in the blood parameters were detected in any case. In fact, Troponin I, the biomarker of choice for the diagnosis of myocardial necrosis [28], exhibited clinically normal values in all swine at the post-therapy timepoints. Therefore, we concluded that the IC approach is safe to deliver S-CDCs, as we have previously verified for other cell-derived therapeutic agents [20].

In the porcine MI-VT model, anteroseptal infarctions with a thin surviving rim of endocardium are normally created. Moreover, surviving strands of myocardium interwoven with fibrotic scar are usually revealed [21]. In accordance with these findings, TTC staining at 4 months in our study revealed transmural fibrous scars of variable extension with anteroseptal location in all cases. Furthermore, a mixture of infarcted/viable tissue, accompanied by ventricular wall thinning, was also present, which is consistent with previous studies of similar porcine MI models [20]. No statistically significant differences between both study groups were detected (neither at a macroscopic nor at a microscopic level).

Despite the absence of significant differences between CON and S-CDCs in the pathological examination, the MRI results seem to indicate that IC treatment with S-CDCs could cause a reduction in infarct size. In recent years, MRI has become an indispensable tool in cardiology. This non-invasive imaging technique allows objective and functional assessment of myocardial tissue [29]. Among the different parameters that can be measured by MRI, infarct size is considered a valid criterion to assess cardiac function [16]. In this regard, in the present study, animals that were submitted to IC S-CDCs treatment exhibited a significantly lower MI at 4 months than the pigs belonging to the CON group.

Other MRI-derived parameters that have been widely described for the assessment of cardiac function include left ventricular volumes and, more importantly, LVEF [16]. In our case, both LVEDVi and LVESVi were lower in the S-CDCs group at the end of the study, indicating limited ventricular remodelling in this study group. On the contrary, LVEF was higher in this group at 4 months, although no statistically significant differences were reached in any of these parameters.

The discrepancy between a reduction in infarct size and the lack of corresponding improvement in cardiac function is a complex issue, which could be largely attributed to the timing of magnetic resonance assessments. Williams et al. (2011) [30] demonstrated that while infarct size reduction could be detected as early as 3 months after therapy, a reduction in chamber volumes was not observed until 6 months post-treatment. This temporal sequence suggests that the recovery of cardiac function may require more time than the 4-month follow-up period used in our study to become evident [30].

### 4.1. Limitations

Despite the promising results of this study, several limitations must be considered. Although the porcine model is the preferred one in preclinical studies comprising the cardiovascular system, careful extrapolation of the results to the clinical scenario needs to be performed. One key limitation is the use of young, healthy pigs, as post-MI VT typically occurs in elderly patients with multiple comorbidities. These conditions were not replicated in our experimental model. Additionally, the pigs that have been used for model creation are still growing, while it is not the case in human patients, who typically experience MI in adulthood. To minimize the influence of weight gain on the interpretation of the results, ventricular volumes have been indexed to BSA [16,19,20]. Moreover, in post-MI patients, beta-blockers are commonly used as effective antiarrhythmic agents [31]. In our study, however, we avoided its usage since this medication could have masked any anti-arrhythmic effect attributed to the administration of S-CDCs.

Regarding post-mortem examination, it needs to be mentioned as a further limitation that a complete analysis of the entire heart has not been performed, since only one representative heart section was analysed microscopically (HE and MT staining), while another heart slice was used for macroscopic evaluation (TTC staining).

Another limitation is the small sample size, which could limit the statistical power of the study and the ability to detect subtle differences between groups [32]. Future studies with larger sample sizes are needed to confirm our findings and further assess the long-term effects of IC S-CDCs on VT inducibility and cardiac function.

### 4.2. Future Directions

The conduction of preclinical studies is a crucial step in the development of new treatments and therapies. However, these studies can sometimes suffer from issues like small sample sizes, lack of reproducibility, or insufficient methodological rigor, which can hinder the progress of medical research. In this context, the establishment of research networks specifically designed to enhance the rigor and robustness of preclinical studies may increase the chances of translating the results to the human patient [33]. Investigating the long-term safety and efficacy of IC S-CDCs in larger sample sizes will help further clarify their role in mitigating VT and improving overall cardiac function after MI. Additionally, clinical trials are needed to evaluate the feasibility and effectiveness of this approach in human patients with arrhythmias after chronic MI. Furthermore, exploring the synergistic potential of S-CDCs in combination with other therapies, such as ICDs or catheter ablation, could offer new avenues for the management of post-MI arrhythmias.

## 5. Conclusions

The IC administration of S-CDCs presents a promising therapeutic approach for mitigating the development of VT following MI, along with a beneficial impact on infarct size. This therapy appears to reduce the % MI in this porcine model. Moreover, the IC administration of S-CDCs in this experimental model of chronic MI appears feasible and safe, as demonstrated by the absence of therapy-related adverse events, changes in ECG, and the lack of significant differences in post-therapy measured blood parameters.

These findings highlight the potential of S-CDCs as a novel therapeutic strategy for patients at risk of developing post-MI VT. Given the absence of significant safety concerns and the positive functional outcomes observed in this preclinical model, S-CDCs may serve as an important tool in the future management of arrhythmias following MI, opening up avenues for further investigation and clinical application in cardiac regenerative medicine.

## Figures and Tables

**Figure 1 biomedicines-13-01043-f001:**
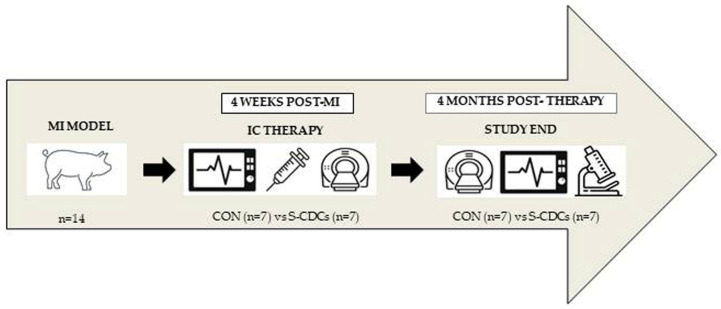
Experimental protocol. MI: myocardial infarction. IC: intracoronary. CON: control group. S-CDCs: cardiosphere-derived cell secretome group.

**Figure 2 biomedicines-13-01043-f002:**
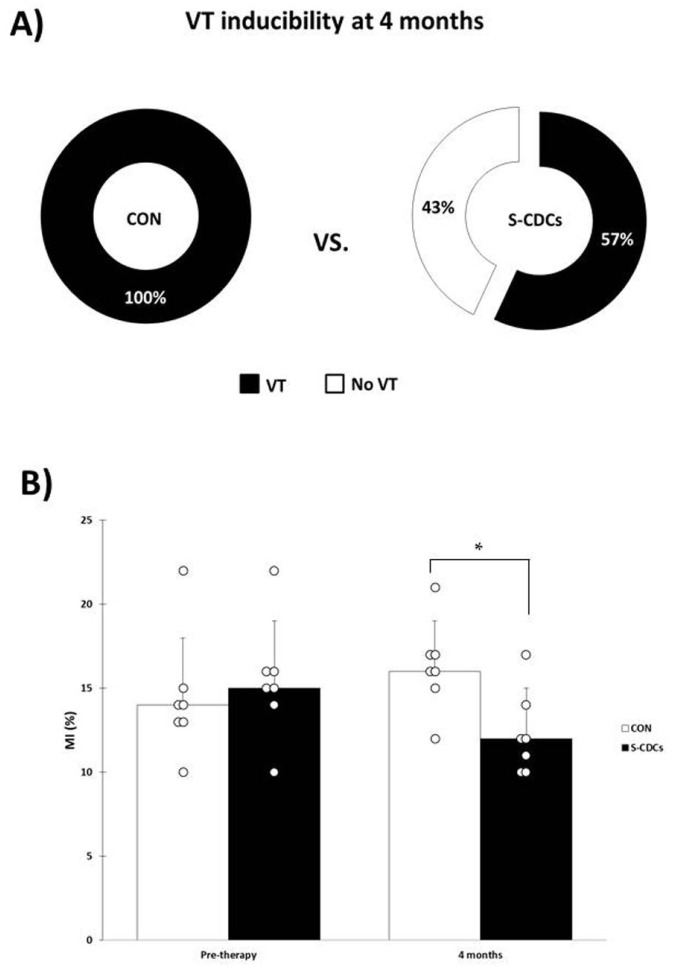
(**A**) VT inducibility rates at four months. VT inducibility is lower in the S-CDCs group compared to CON (*p* = 0.05). (**B**) Infarct size (MI) in the CON and S-CDCs groups before therapy and four months later. * denotes statistical significance (*p* < 0.05). At four months MI is significantly lower in the S-CDCs group (*p* = 0.03).

**Figure 3 biomedicines-13-01043-f003:**
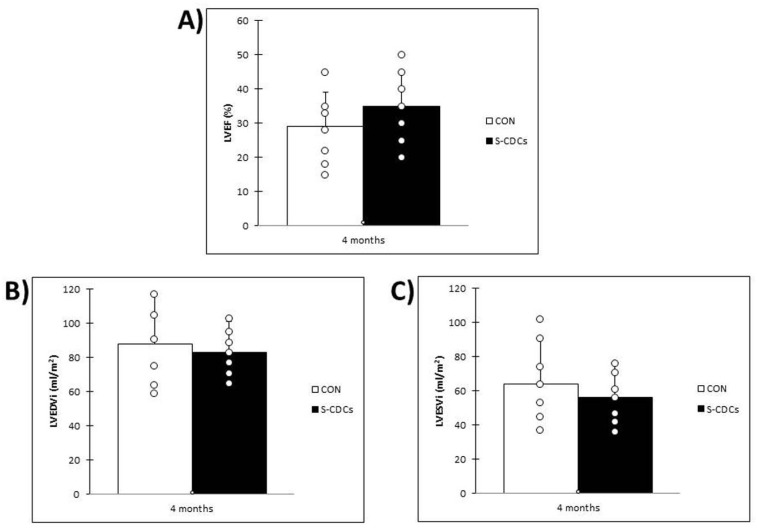
Data obtained in MRI studies four months after IC therapy. (**A**) Left ventricular ejection fraction (LVEF) in the CON and S-CDCs groups. No significant differences were detected between groups (*p* = 0.25). (**B**) Indexed left ventricular end diastolic volume (LVEDVi) in the CON and S-CDCs groups (*p* = 0.66). No significant differences were detected between groups. (**C**) Indexed left ventricular end systolic volume (LVESVi) in the CON and S-CDCs groups. No significant differences were detected between groups (*p* = 0.41).

**Figure 4 biomedicines-13-01043-f004:**
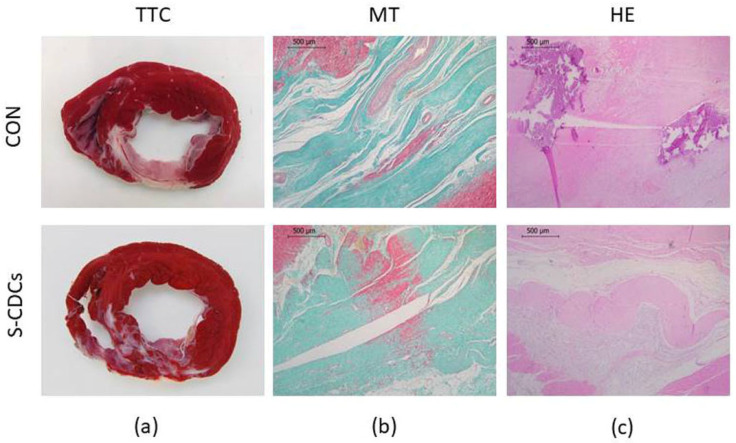
Macroscopical and histopathological appearance of MI in both groups. (**a**) TTC staining showing the extension of the infarct area in CON and S-CDCs groups. Representative images of the infarct area from both study groups (**b**) MT staining. (**c**) HE staining.

**Table 1 biomedicines-13-01043-t001:** MRI-derived parameters before and 4 months after therapy. Data presented as mean ± standard deviation. Intragroup comparisons revealed no statistically significant differences in any measured parameter. CON: control group. S-CDCs: cardiosphere-derived cell secretome group. LVEF: left ventricular ejection fraction. % MI: infarct size as a percentage of left ventricle. LVEDVi: left ventricular indexed end-diastolic volume. LVESVi: left ventricular indexed end-systolic volume.

Study Group	CON			S-CDCs		
Timepoint	Pre-Therapy	4 Months Post-Therapy	*p*-Value	Pre-Therapy	4 Months Post-Therapy	*p*-Value
**LVEF (%)**	32 ± 6	29 ± 10	0.27	35 ± 5	35 ± 10	0.79
**LVEDVi (mL/m^2^)**	88 ± 19	88 ± 29	0.87	83 ± 14	83 ± 18	0.50
**LVESVi (mL/m^2^)**	66 ± 15	64 ± 27	0.80	58 ± 13	56 ± 20	0.35
**MI (%)**	14 ± 4	16 ± 3	0.20	15 ± 4	12 ± 3	0.18

## Data Availability

Datasets analysed or generated during the study are available from the corresponding author on reasonable request.

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
