# Peer review of "Intracoronary-Cardiosphere-Derived Cell Secretome Therapy: Effects on Ventricular Tachycardia Inducibility and Cardiac Function in a Swine Model"

_biomedicines, 2025, doi:10.3390/biomedicines13051043_

Round 1
Reviewer 1 Report (Previous Reviewer 2)
Comments and Suggestions for Authors
I reviewed the article entitled Intracoronary cardio sphere derived cell secretome therapy: Effects on ventricular tachycardia inducibility and cardiac function in a swine model. The topic of the study is original and provides significant information to the literature. My comments:
1- The authors mention catheter ablation in post-VT in the introduction. However, before doing so, they should clarify this issue and expand the introduction. How many groups are post MI VT divided into? What are early and late stage VT? How are they distinguished? What are the treatment options? These should be discussed in detail in the introduction. How many days after MI should VT be considered permanent?
2- Introduction, 3rd paragraph, CDC should be detailed defined. What is CDC? How are they obtained? Please add.
3- Why the authors did not perform power analysis before the study started? Sample size should be detected with the power analysis? Also, in current form, with using power analysis, what is the power of the study? please Express.
4- ın the table S1, authors stated that the reference value of the creatinine is <2.3??? is it true? I think that the reference of creatinine is <1.2?? please check.
5- Tablo S1 should include the detailed p values.
6- ın statistical analysis, authors stated that Differences between groups were calculated using the Kruskal-Wallis and Mann-Whitney U tests. Kruskal Wallis is used in comparison of the more than 2 independent groups. However, I did not find any data in the text that should be performed kruskal Wallis test. Also, Table S1, there are more than 2 repeated measures. Comparison should be made with Firedman. However, I did not any sentence that describe this. Please, help from a statistician.
7- Was there any difference between CON and SCD groups regarding baseline EF, EDVi, Esv, and MI%. Please compare with the mann Whitney U test. It seems that pre-therapy EF is lower and , EDVi and ESVi are higher in CON group. If there is a difference between the groups in this respect, could this have affected the results?
Sincerely.
Comments on the Quality of English LanguageEnglish language is fine.
Author Response
First of all, we would like to thank the reviewer for the valuable comments and suggestions that helped us improve the clarity and relevance of this manuscript. We have modified the manuscript according to the reviewer’s comments. All changes made in the text are highlighted. Here, we provide a detailed list of answers to the specific comments made by the reviewer. The original reviewer’s comments are reproduced in italics and bold before our responses.
Reviewer :
I reviewed the article entitled Intracoronary cardio sphere derived cell secretome therapy: Effects on ventricular tachycardia inducibility and cardiac function in a swine model. The topic of the study is original and provides significant information to the literature. My comments:
1- The authors mention catheter ablation in post-VT in the introduction. However, before doing so, they should clarify this issue and expand the introduction. How many groups are post MI VT divided into? What are early and late stage VT? How are they distinguished? What are the treatment options? These should be discussed in detail in the introduction. How many days after MI should VT be considered permanent?
According to the reviewer´s comment the Introduction section has been modified and expanded as follows:
“Post-infarction VT is categorized as early or late depending on the time of onset: Early VT, occurring within 48-72 hours post-MI, is generally transient and associated with acute ischemia, reperfusion injury, or metabolic imbalances, with treatment aimed at correcting reversible factors. In contrast, late VT, developing after 72 hours (often weeks to months later), is typically caused by re-entry circuits established by slow conduction channels within the heterogeneous tissue (HT) immersed in the non-excitable dense scar. VT occurring more than 40 days post-MI is generally considered permanent, as it reflects underlying scar-related substrate. The management of late VT frequently involves implantable cardioverter-defibrillators and catheter ablation among others [4].
Catheter ablation is a usual technique that is carried out in up to 20% of the patients for targeting this arrhythmogenic substrate [5]. Emerging therapeutic strategies are exploring the modulation of this process through biological agents, offering a promising frontier in the treatment of post-MI arrhythmias [6]”.
A new reference has been added to the manuscript:
- Al-Khatib S.M., Stevenson W.G., Ackerman M.J., Bryant W.J., Callans D.J., Curtis A.B., Deal B.J., Dickfeld T., Field M.E., Fonarow G.C., Gillis A.M., Granger C.B., Hammill S.C., Hlatky M.A., Joglar J.A., Kay G.N., Matlock D.D., Myerburg R.J., Page R.L. 2017 AHA/ACC/HRS Guideline for Management of Patients With Ventricular Arrhythmias and the Prevention of Sudden Cardiac Death: Executive Summary: A Report of the American College of Cardiology/American Heart Association Task Force on Clinical Practice Guidelines and the Heart Rhythm Society. Circulation. 2018; 138(13):e210-e271. doi: 10.1161/CIR.0000000000000548. Erratum in: Circulation. 2018; 138(13):e415-e418. doi: 10.1161/CIR.0000000000000615.
2- Introduction, 3rd paragraph, CDC should be detailed defined. What is CDC? How are they obtained? Please add.
Cardiosphere-Derived Cells (CDCs) are a heterogeneous population of cardiac progenitor cells typically isolated from small heart tissue biopsies. They are generated through a multistep culture process involving enzymatic digestion, formation of cardiospheres, and subsequent adherent culture. CDCs possess a distinct surface marker profile (CD105⁺, CD90⁺, c-kit⁻/low) and exert reparative effects primarily via paracrine signaling, contributing to angiogenesis, reduced fibrosis, and enhanced cardiomyocyte survival.
This information has been added in the revised version of the manuscript as follows:
“Among these emerging strategies, cardiosphere-derived cells (CDCs), a heterogeneous population of cardiac progenitor cells typically isolated from heart tissue biopsies, have shown significant promise in preclinical and clinical settings. These cells, generated through a multistep culture process involving enzymatic digestion, formation of cardiospheres, and subsequent adherent culture, are able to reduce scar mass, enhance viable myocardial tissue, and prevent adverse remodelling, thereby offering a potential way of halting the development of new arrhythmogenic substrates [7-11]. However, recent findings suggest that the therapeutic effects of CDCs are largely mediated by the release of their secretome (S-CDCs) [12, 13]”.
3- Why the authors did not perform power analysis before the study started? Sample size should be detected with the power analysis? Also, in current form, with using power analysis, what is the power of the study? Please Express.
Sample size was estimated for the original primary endpoint, which was improvement of EF as measured by Magnetic Resonance, based on prior results observed in our institution. Using the G*Power software (https://www.gpower.hhu.de), with a power of 85% and an alpha error of 5% a sample size of n=7 (per group) was considered enough to detect a 10% difference in EF between groups.
A post hoc power analysis has been conducted using the observed effect size for VT inducibility, the sample size (n=7 per group), and a significance level of 0.05. The resulting power was estimated to be 67%, suggesting limited power to detect the observed effect.
4- ın the table S1, authors stated that the reference value of the creatinine is <2.3??? is it true? I think that the reference of creatinine is <1.2?? please check.
According to the reviewer´s comment, we have checked reference value of creatinine. The reference range for creatinine in swine, as established in our laboratory, is 0.8 to 2.3 mg/dL. Additionally, the literature reports reference values as high as 2.7 mg/dL (Kaneko et al., 2008).
- Kaneko, J.J., Harvey, J.W. and Bruss, M. (2008) Clinical Biochemistry of Domestic Animals. 6th Edition, Elsevier
- msdvetmanual.com/reference-values-and-conversion-tables/reference-guides/serum-biochemical-analysis-reference-ranges
5- Table S1 should include the detailed p values.
Thank you for your comment. The missing p-values have been added to Table S1.
6- ın statistical analysis, authors stated that Differences between groups were calculated using the Kruskal-Wallis and Mann-Whitney U tests. Kruskal Wallis is used in comparison of the more than 2 independent groups. However, I did not find any data in the text that should be performed kruskal Wallis test. Also, Table S1, there are more than 2 repeated measures. Comparison should be made with Firedman. However, I did not any sentence that describe this. Please, help from a statistician.
For statistical analysis, authors used Mann-Whitney U test for comparisons between CON and S-CDCs group. Wilcoxon test was used to compare pre- and 4 months post-therapy timepoints. We agree with the reviewer and Friedman test has been therefore included to compare repeated measures (Table S1).
This information was clarified in the Material and Methods section as follows:
“Obtained data are presented as means ± standard deviations. Differences between groups were calculated using the Mann-Whitney U tests. The Wilcoxon paired sample test was used to perform intragroup comparisons (pre- and 4 months post-therapy timepoints). For repeated measures the Friedman test was performed. Binary data were analysed by performing a chi-square test. Values of p < 0.05 were considered statistically significant. All calculations were carried out using the SPSS 18.0 statistical package for Windows (SPSS Inc., Chicago, IL, USA)”.
7- Was there any difference between CON and SCD groups regarding baseline EF, EDVi, Esv, and MI%. Please compare with the mann Whitney U test. It seems that pre-therapy EF is lower and, EDVi and ESVi are higher in CON group. If there is a difference between the groups in this respect, could this have affected the results?
Thank you for your comment. Although at baseline EF was lower and EDVi and ESVi were higher in CON group, no statistically significant differences were detected between groups, as stated in the manuscript:
“Regarding the parameters evaluated by MRI (Table 1), before therapy, no significant differences between CON and S-CDCs groups were observed in LVEF (32±6 % vs. 35±5 %, p=0.30, respectively), % MI (14±4 vs. 15±4 %, p=0.56, respectively), LVEDVi (88±19 ml/m2 vs. 83±14 ml/m2, p=0.52, respectively) and LVESVi (66±15 ml/m2 vs. 58±13 ml/m2, p=0.10, respectively)”.
If our groups had shown statistically significant differences at baseline, any observed differences at four months post-therapy could have reflected those initial disparities rather than the true effect of S-CDC therapy. This would have introduced potential bias and may have led to misleading conclusions regarding the treatment’s efficacy.
Reviewer 2 Report (Previous Reviewer 3)
Comments and Suggestions for Authors
Hello,
Thanks for sharing manuscript titled "Intracoronary cardiosphere derived cell secretome therapy: Effects on ventricular tachycardia inducibility and cardiac function in a swine model' to this journal.
It is a well written and scientifically conducted research worthy of publication. I have some minor issues that need to be clarified -
- How was the "healthy" status of the pigs established at baseline to avoid confounding due to baseline health or diseased status of their cardiac status ?
- Why was the LAD occlusion done after the second diagonal and not first diagonal or septal and what is the scientific basis of 150 min of occlusion?
Thanks
Author Response
First of all, we would like to thank the reviewer for the valuable comments and suggestions that helped us improve the clarity and relevance of this manuscript. We have modified the manuscript according to the reviewer’s comments. All changes made in the text are highlighted. Here, we provide a detailed list of answers to the specific comments made by the reviewer. The original reviewer’s comments are reproduced in italics and bold before our responses.
Reviewer:
Hello,
Thanks for sharing manuscript titled "Intracoronary cardiosphere derived cell secretome therapy: Effects on ventricular tachycardia inducibility and cardiac function in a swine model' to this journal.
It is a well written and scientifically conducted research worthy of publication. I have some minor issues that need to be clarified -
- How was the "healthy" status of the pigs established at baseline to avoid confounding due to baseline health or diseased status of their cardiac status?
In order to ensure animal health and welfare, all subjects arrived at our institution at least two weeks prior to the first scheduled intervention. This allowed for a thorough clinical examination and a quarantine period aimed at detecting any subclinical condition. Additionally, baseline ECG recording and basic blood tests (including for example troponin I levels) were performed before the animals were included in the protocol.
- Why was the LAD occlusion done after the second diagonal and not first diagonal or septal and what is the scientific basis of 150 min of occlusion?
The LAD occlusion was performed distal to the second diagonal branch to create a consistent and clinically relevant myocardial infarction (MI) that effectively models the development of ventricular tachycardia (VT). This particular occlusion site allows for a consistent infarct size and location, promoting the development of the electrophysiological substrate necessary for VT (Sasano et al., 2009).
The decision to use a 150-minute balloon occlusion is based on its established use in porcine MI models, which are known to consistently induce VT. Previous studies have shown that this duration leads to VT inducibility in approximately 90% of the animals within 4–5 weeks post-MI. In our study, this approach resulted in a 100% VT inducibility rate at 4 weeks, confirming the model’s reliability and effectiveness. This high rate of inducibility highlights the model's suitability for evaluating therapeutic interventions like S-CDCs aimed at reducing post-MI arrhythmias (Sasano et al., 2009; Tschabrunn et al., 2009).
- Sasano T.; Kelemen K., Greener I.D.; Donahue J.K. Ventricular tachycardia from the healed myocardial infarction scar: validation of an animal model and utility of gene therapy. Heart Rhythm. 2009; 6(8 Suppl):S91-7. doi: 10.1016/j.hrthm.2009.03.048.
- Tschabrunn C.M.; Roujol S.; Nezafat R.; Faulkner-Jones B., Buxton A.E.; Josephson M.E.; Anter E. A swine model of infarct-related reentrant ventricular tachycardia: Electroanatomic, magnetic resonance, and histopathological characterization. Heart Rhythm. 2016; 13(1):262-73. doi: 10.1016/j.hrthm.2015.07.030.
This explanation is reflected in the discussion section of the manuscript as follows:
“In our case, we utilized a porcine MI model induced by 150 minutes of balloon occlusion. This model is widely regarded for its ability to reliably induce VT, with 90% of animals typically demonstrating VT inducibility within 4-5 weeks post-MI [21]. In our study, we achieved a 100% inducibility rate at the 4 weeks follow-up, which aligns with previously published data from similar models [22]. This high inducibility rate reflects the effectiveness of the used model in reproducing the electrophysiological substrate for VT, providing an appropriate platform to assess the antiarrhythmic effects of S-CDCs”.
Reviewer 3 Report (New Reviewer)
Comments and Suggestions for Authors
This article describes an essential in vivo study to test the feasibility and efficacy of cardiosphere derived cell secretome (S-CDCs) in preventing ventricle tachycardia (VT) following myocardial infarction (MI) using a well-sized porcine cohort. This cohort comprising 14 animals (7 in the control group and 7 in the treatment group) represents an appropriate study design needed to not only acquire preclinical application information of S-CDCs but also serve as a guide for efficacy testing for other important cardiovascular therapeutics in porcine model. In this study, the S-CDCs administration showed a curative effect for VT, although the difference was insignificant. However, S-CDCs treatment has led to substantial MI recovery, which reached significance with p value 0.03. The only concern is that MI recovery is not reflected in the representative image in Figure 4, panel TTC. The authors are therefore suggested to add a representative image in Figure 4.
Author Response
First of all, we would like to thank the reviewer for the valuable comments and suggestions that helped us improve the clarity and relevance of this manuscript. We have modified the manuscript according to the reviewer’s comments. All changes made in the text are highlighted. Here, we provide a detailed list of answers to the specific comments made by the reviewer. The original reviewer’s comments are reproduced in italics and bold before our responses.
Reviewer:
This article describes an essential in vivo study to test the feasibility and efficacy of cardiosphere derived cell secretome (S-CDCs) in preventing ventricle tachycardia (VT) following myocardial infarction (MI) using a well-sized porcine cohort. This cohort comprising 14 animals (7 in the control group and 7 in the treatment group) represents an appropriate study design needed to not only acquire preclinical application information of S-CDCs but also serve as a guide for efficacy testing for other important cardiovascular therapeutics in porcine model. In this study, the S-CDCs administration showed a curative effect for VT, although the difference was insignificant. However, S-CDCs treatment has led to substantial MI recovery, which reached significance with p value 0.03. The only concern is that MI recovery is not reflected in the representative image in Figure 4, panel TTC. The authors are therefore suggested to add a representative image in Figure 4.
According to the reviewer´s comment, a representative image has been included in Figure 4 (panel TTC) for the S-CDCs group as follows:

Round 2
Reviewer 1 Report (Previous Reviewer 2)
Comments and Suggestions for Authors
Authors responded in a satisfactory way. Thank you.
This manuscript is a resubmission of an earlier submission. The following is a list of the peer review reports and author responses from that submission.
Round 1
Reviewer 1 Report
Comments and Suggestions for Authors
Intracoronary cardiosphere derived cell secretome therapy: Effects on ventricular tachycardia inducibility and cardiac function in a swine model
Summary:
This is randomized blinded preclinical animal study using 14 pigs who underwent endovascular MI model creation and received intracoronary saline or cardiosphere derived cell secretome (S-CDC) therapy to assess VT inducibility. The authors concluded that pigs treated with S-CDCs had lower VT inducibility rates and percentage of myocardial ischemia. Authors conclude that those treated with S-CDCs had lower incidence of inducible VT and % MI compared to the control-treated pigs. This is a well described and well-designed study showing potential clinical application.
Major
1. Although it is stated in parenthesis that findings are non-significant, results in the abstract can be misleading by stating that LVEF was higher, and LV volumes were lower between both groups. Would clearly specify that while there was a numerical trend towards improvement, it was not statistically significant.
2. How are the authors explaining the discrepancy between changes in %MI on MRI but no changes in histological data regarding scar extent and location as well as macroscopic lesions?
3. In the discussion (line 252-253 and again 266), authors hypothesize that S-CDCs help modify EP properties and VT substrate via anti-fibrotic effect and ventricular modeling effect. However, the histological data is not consistent with this (no change in TTC staining or histological data and no change in remodeling on MRI except for % MI). Would need further explanation for these discrepancies if this is the hypothesis.
4. Would be careful about overstating several findings. For example, inaccurate statement needs correction (line 215, 291-292, 337-338) stating MRI results did not show improvement in cardiac function (LVEF had numerical improvement but not statistical significance). While infarct size was significantly improved, it is not a marker of cardiac function particularly if LVEF has not changed. Line 299-302 need to specify that findings were not statistically significant.
Minor:
1. Would clarify what MI means (line 193).
Author Response
Authors’ responses to Reviewer 1 (Round 1)
First of all, we would like to thank the reviewer for the valuable comments and suggestions that helped us improve the clarity and relevance of this manuscript. We have modified the manuscript according to the reviewer’s comments. All changes made in the text are highlighted. Here, we provide a detailed list of answers to the specific comments made by the reviewer. The original reviewer’s comments are reproduced in italics and bold before our responses.
Reviewer 1:
Summary:
This is randomized blinded preclinical animal study using 14 pigs who underwent endovascular MI model creation and received intracoronary saline or cardiosphere derived cell secretome (S-CDC) therapy to assess VT inducibility. The authors concluded that pigs treated with S-CDCs had lower VT inducibility rates and percentage of myocardial ischemia. Authors conclude that those treated with S-CDCs had lower incidence of inducible VT and % MI compared to the control-treated pigs. This is a well described and well-designed study showing potential clinical application.
Major
- Although it is stated in parenthesis that findings are non-significant, results in the abstract can be misleading by stating that LVEF was higher, and LV volumes were lower between both groups. Would clearly specify that while there was a numerical trend towards improvement, it was not statistically significant.
As reviewer suggests, we have modified the Abstract as follows:
“Moreover, LVEF was higher (35±10 % versus 29±10 %, p=NS) and ventricular volumes were lower (83±18 ml/m2 and 56±20 ml/m2 versus 88±29 ml/m2 and 64±20 ml/m2, p=NS) in this group, although no statistical significant differences between groups were observed”.
- How are the authors explaining the discrepancy between changes in %MI on MRI but no changes in histological data regarding scar extent and location as well as macroscopic lesions?
The authors assume that the discrepancy between the changes in %MI observed on MRI and the absence of corresponding changes in histological data regarding infarct scar and macroscopic lesions may be attributed to the use of a single heart slice for histopathological examination, while another single section was utilized for TTC staining. We strongly believe that analysing additional sections of the heart would have provided a more thorough representation of the infarct. Consequently, the comparison between MRI data and histology would likely have resulted in more consistent findings. The lack of a complete heart analysis has been included as a further limitation of the study:
“Regarding post-mortem examination, it needs to be mentioned as a further limitation, that a complete analysis of the entire heart has not been performed, since only one representative heart section was analyzed microscopically (HE and MT staining), while another heart slice was used for macroscopic evaluation (TTC staining)”.
Moreover, in order to avoid misunderstanding, we have modified related information in the Materials and Methods section as follows:
“Samples from normal tissue (NT, normal myocytes and the absence of fibrosis) and infarcted area were collected from one of the remaining slices for posterior histopathological analysis by means of haematoxylin-eosin (HE) and Masson’s trichrome (MT) staining”.
- In the discussion (line 252-253 and again 266), authors hypothesize that S-CDCs help modify EP properties and VT substrate via anti-fibrotic effect and ventricular modeling effect. However, the histological data is not consistent with this (no change in TTC staining or histological data and no change in remodeling on MRI except for % MI). Would need further explanation for these discrepancies if this is the hypothesis.
Thank you for your comment. As mentioned in the introduction of the manuscript, our starting hypothesis was that the IC administration of S-CDCs might reduce the inducibility of sustained ventricular arrhythmias (without specifying whether this capacity could be attributed to an anti-fibrotic effect or an impact on ventricular remodeling). Accordingly, we have corrected our statement regarding hypothesis in the discussion, as follows:
“In the present study, we hypothesized that S-CDCs could help modify the electrophysiological properties of the infarcted tissue, thereby reducing the substrate for VT”.
“As in the case of transendocardial delivery, we assume that VT is rendered non-inducible by the S-CDCs via improving conduction in areas of isolated potentials [1]”.
- Would be careful about overstating several findings. For example, inaccurate statement needs correction (line 215, 291-292, 337-338) stating MRI results did not show improvement in cardiac function (LVEF had numerical improvement but not statistical significance). While infarct size was significantly improved, it is not a marker of cardiac function particularly if LVEF has not changed. Line 299-302 need to specify that findings were not statistically significant.
Following the reviewer’s suggestion, we have corrected inaccurate statements in the new revised version of the manuscript as follows:
- Discussion section:
“Furthermore, MRI analysis demonstrated an improvement in infarct size in the S-CDCs group at that timepoint, suggesting a beneficial therapeutic effect of S-CDCs on the damaged myocardial tissue”.
“Despite the absence of significant differences between CON and S-CDCs in the pathological examination, the MRI results seem to indicate that IC treatment with S-CDCs could cause a reduction in infarct size”.
- Conclusions section:
“The IC administration of S-CDCs presents a promising therapeutic approach for mitigating the development of VT following MI, along with a beneficial impact on infarct size”.
Minor:
- Would clarify what MI means (line 193).
According to the reviewer, it has been clarified what MI means:
“Infarct size, however, was significantly lower in the group treated with S-CDCs (12±3 % vs. 16±3 %; p=0.03) (Figure 2 B)”.
Reviewer 2 Report
Comments and Suggestions for Authors
Dear editor
I reviewed the article entitled Intracoronary cardio sphere derived cell secretome therapy: Effects on ventricular tachycardia inducibility and cardiac function in a swine model. The topic of the study is original and provides significant information to the literature. My comments:
1- Authors stated that “myocardial infarction (MI) carries a high risk of sudden death primarily due to the development of ventricular tachycardia”. Please add the current proportion of sudden death during MI and late term post MI with the literature information. Also, what percentage of VTs after MI require ablation? Please add.
2- In the results section, please indicate the certain p values instead of NS.
3- Please add the certain p values to the figure 2 for each variables.
4- Please compare the pre and post variables intragroups, for example; LVEF in SCD groups basal and 4. Months; LVEF in CON basal and 4. Months. Compare each variable intragroups for both groups.
Author Response
Authors’ responses to Reviewer 2 (Round 1)
First of all, we would like to thank the reviewer for the valuable comments and suggestions that helped us improve the clarity and relevance of this manuscript. We have modified the manuscript according to the reviewer’s comments. All changes made in the text are highlighted. Here, we provide a detailed list of answers to the specific comments made by the reviewer. The original reviewer’s comments are reproduced in italics and bold before our responses.
Reviewer 2:
Dear editor
I reviewed the article entitled Intracoronary cardio sphere derived cell secretome therapy: Effects on ventricular tachycardia inducibility and cardiac function in a swine model. The topic of the study is original and provides significant information to the literature. My comments:
- Authors stated that “myocardial infarction (MI) carries a high risk of sudden death primarily due to the development of ventricular tachycardia”. Please add the current proportion of sudden death during MI and late term post MI with the literature information. Also, what percentage of VTs after MI require ablation? Please add.
Sudden cardiac death (SCD) is common during and after a MI, with 29.7% of patients experiencing sudden death within the first month (Curtain et al., 2024) and SCD accounting for 24% to 40% of post-MI mortality (Zaman and Kovoor, 2014).
Approximately 10-20% of patients with post-MI VT may require catheter ablation, especially when VT is frequent, symptomatic, or refractory to medical therapy (Dukkipati et al., 2017).
As suggested by the reviewer, the proportion of sudden death related to MI and percentage of VTs after MI that require ablation have been added to the manuscript as follows:
“Myocardial infarction (MI) carries a high risk of sudden cardiac death (SCD) primarily due to the development of ventricular tachycardia (VT) [1]. Specifically, almost 30% of the patients suffer from SCD within the first month, being responsible for up to 40% of post-MI mortality [2, 3].
Most post-infarction VTs originate in re-entry circuits established by slow conduction channels within the heterogeneous tissue (HT) immersed in the non-excitable dense scar. While catheter ablation is a usual technique that is carried out in up to 20% of the patients for targeting this arrhythmogenic substrate [4], emerging therapeutic strategies are exploring the modulation of this process through biological agents, offering a promising frontier in the treatment of post-MI arrhythmias [5]”.
To this respect, the following references have been included in the revised version of the manuscript:
- Curtain J.P., Pfeffer M.A., Braunwald E., Claggett B.L., Granger C.B., Køber L., Lewis E.F., Maggioni A.P., Mann D.L., Rouleau J.L., Solomon S.D., Steg P.G., Finn P.V., Fernandez A., Jering K.S., McMurray J.J.V. Rates of Sudden Death After Myocardial Infarction—Insights From the VALIANT and PARADISE-MI Trials. JAMA Cardiol. 2024; 9(10):928–933. doi:10.1001/jamacardio.2024.2356.
- Zaman S., Kovoor P. Sudden cardiac death early after myocardial infarction: pathogenesis, risk stratification, and primary prevention. Circulation. 2014; 129(23):2426-35. doi: 10.1161/CIRCULATIONAHA.113.007497.
- Dukkipati S.R., Koruth J.S., Choudry S., Miller M.A., Whang W., Reddy V.Y. Catheter Ablation of Ventricular Tachycardia in Structural Heart Disease: Indications, Strategies, and Outcomes-Part II. J Am Coll Cardiol. 2017; 70(23):2924-2941. doi: 10.1016/j.jacc.2017.10.030.
- In the results section, please indicate the certain p values instead of NS.
According to the reviewer’s comment, certain p-values have been included in the results section.
- Please add the certain p values to the figure 2 for each variables.
In accordance with reviewer’s comment, certain p-values have been added to figure 2.
4- Please compare the pre and post variables intragroups, for example; LVEF in SCD groups basal and 4. Months; LVEF in CON basal and 4. Months. Compare each variable intragroups for both groups.
Thank you for your comment. We have included a supplementary table (Table S2) with pre- and 4 month post-therapy data and performed intragroup comparisons. No statistically significant differences were observed.
|
Group |
CON |
S-CDCs |
||||
|
Timepoint |
Pre-therapy |
4 Months Post-therapy |
p-value |
Pre-therapy |
4 Months Post-therapy |
p-value |
|
LVEF (%) |
32±6 |
29±10 |
0.27 |
35±5 |
35±10 |
0.79 |
|
EDVi (ml/m2) |
88±19 |
88±29 |
0.87 |
83±14 |
83±18 |
0.50 |
|
ESVi (ml/m2) |
66±15 |
64±27 |
0.80 |
58±13 |
56±20 |
0.35 |
|
MI (%) |
14±4 |
16±3 |
0.20 |
15±4 |
12±3 |
0.18 |
Reviewer 3 Report
Comments and Suggestions for Authors
Hello
Thanks for submitting this manuscript titled "Intracoronary cardiosphere derived cell secretome therapy: Effects on ventricular tachycardia inducibility and cardiac function in a swine model". It is a well conducted study and well drafted paper.
Small clarifications need to mentioned -
1.The swine models underwent Infarct induction by which method and what was the method of randomisation for control vs CDC-S arms
2. what were the "standard care" offered in infarct models as in the real world settings beta-blockers which have good anti-arrythmic property are often administered in real world setting ? if the same wasn't assessed then this needs to be mentioned in the limitation of the study.
Thanks
Author Response
Authors’ responses to Reviewer 3 (Round 1)
First of all, we would like to thank the reviewer for the valuable comments and suggestions that helped us improve the clarity and relevance of this manuscript. We have modified the manuscript according to the reviewer’s comments. All changes made in the text are highlighted. Here, we provide a detailed list of answers to the specific comments made by the reviewer. The original reviewer’s comments are reproduced in italics and bold before our responses.
Reviewer 3:
Hello
Thanks for submitting this manuscript titled "Intracoronary cardiosphere derived cell secretome therapy: Effects on ventricular tachycardia inducibility and cardiac function in a swine model". It is a well conducted study and well drafted paper.
Small clarifications need to mentioned -
- The swine models underwent Infarct induction by which method and what was the method of randomisation for control vs CDC-S arms
Thank you for your comment. As stated in the Material and Methods section (2.2. Infarct induction) an endovascular infarct induction model was used: A percutaneous coronary angioplasty balloon of appropriate diameter was used to transiently occlude the left anterior descending coronary artery (distal to the second diagonal branch) via a percutaneous femoral approach. Balloon occlusion was maintained during 150 minutes followed by reperfusion. Possible ventricular fibrillation episodes during infarct induction were treated by manual chest compressions and 200 J biphasic defibrillation shocks as well as medication when needed.
With respect to the method of randomization, group allocation was carried out by an independent researcher, so that group assignment was not known to the research team performing the therapies. This statement has been included in the manuscript as follows:
“Animals included in the study were blindly allocated (randomization was carried out by an independent researcher using random number generation) to one of the following groups: Control (CON) or S-CDCs group”.
- What were the "standard care" offered in infarct models as in the real world settings beta-blockers which have good anti-arrhythmic property are often administered in real world setting? If the same wasn't assessed then this needs to be mentioned in the limitation of the study.
Thanks
Regarding “standard care” offered in the porcine infarction model, we used the following medications:
Before infarct induction, oral amiodarone (400 mg) was administered from 5 days prior to the infarction until 3 days after it. Acetylsalicylic acid (500 mg) was given from 24 hours before the model creation until euthanasia. Lidocaine was continuously infused at a rate of 1 mg/kg/h during infarction.
Although beta-blockers have good antiarrhythmic properties and are therefore often administered in the real world setting, in our case we preferred to avoid beta-blocker administration, since this medication could have masked any anti-arrhythmic effects attributed to S-CDCs therapy. As suggested by the reviewer, this issue has been addressed in the limitation section as follows:
“Moreover, in post-MI patients beta-blockers are commonly used as effective antiarrhythmic agents [32]. In our study, however, we avoided it´s usage since this medication could have masked any anti-arrhythmic effect attributed to S-CDCs administration”.
An additional reference has been included in the manuscript:
“32. Görenek B.; Çalık A.N.; Kepez A.; Öz A.; Özmen Ç.; Sinan Ü.Y.; Yontar O.C.; Yıldırım Ç. Antiarrhythmic Properties of Beta Blockers: Focus on Nebivolol. Int J Cardiovasc Acad. 2024;10(2):22-30. doi:10.4274/ijca.2024.85057”.